# MORDAL: AUTOMATED PRETRAINED MODEL SELECTION FOR VISION LANGUAGE MODELS

**Shiqi He, Insu Jang & Mosharaf Chowdhury**
Department of Computer Science and Engineering
University of Michigan
Ann Arbor, MI 48109, USA
{shiqihe,insujang,mosharaf}@umich.edu

## ABSTRACT

Incorporating multiple modalities into large language models (LLMs) is a powerful way to enhance their understanding of non-textual data, enabling them to perform multimodal tasks. Vision language models (VLMs) form the fastest growing category of multimodal models because of their many practical use cases, including in healthcare, robotics, and accessibility. Unfortunately, even though different VLMs in the literature demonstrate impressive visual capabilities in different benchmarks, they are handcrafted by human experts; there is no automated framework to create task-specific multimodal models.

We introduce Mordal [1], an automated multimodal model search framework that efficiently finds the best VLM for a user-defined task without manual intervention. Mordal achieves this both by reducing the number of candidates to consider during the search process and by minimizing the time required to evaluate each remaining candidate. Our evaluation shows that Mordal can find the best VLM for a given problem using $8.9\times$–$11.6\times$ lower GPU hours than grid search. We have also discovered that Mordal achieves about 69% higher weighted Kendall's $\tau$ on average than the state-of-the-art model selection method across diverse tasks.

## 1 INTRODUCTION

Vision Language Models (VLMs) bridge the gap between visual and language understanding, rising as the dominant approach to solving visual information-based tasks. Notably, GPT-4o Hurst et al. (2024), a large-scale multimodal language model, demonstrates impressive vision reasoning capabilities by taking images as input and generating detailed natural language descriptions. Although the technical details behind GPT-4o remain undisclosed, researchers have proposed a number of publicly available VLMs (e.g., LLaVA Liu et al. (2023a), InternVL Chen et al. (2024b) and Qwen-VL Bai et al. (2023)) that aim to match GPT-4o's capabilities. Many of these open-source VLMs share a similar ar-

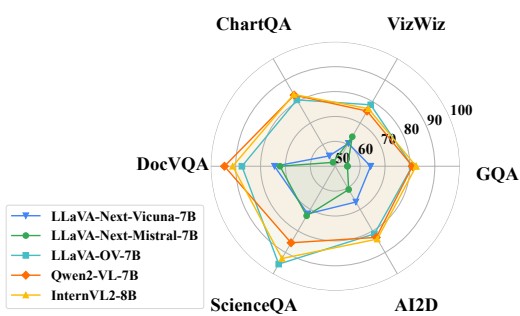

Figure 1: Benchmark performance of five latest *open-source* VLMs on six multimodal tasks.

chitecture, in which a feature projector converts the image embeddings generated by a vision encoder and feeds it to a large language model (LLM) along with the text embeddings.

To construct and train a VLM, the common approach starts from selecting an appropriate *pretrained* vision encoder and LLM. Thanks to the ever-growing ecosystem like HuggingFace Face (2025), developers are able to choose from countless pretrained models for their own VLMs. Unfortunately, despite the great number of available models, it is difficult to determine which pretrained models (i.e., vision encoders and LLMs) are the most appropriate ones. Given a user-specific downstream

---

[1]Mordal is available at https://github.com/SymbioticLab/Mordal.

task, it is unclear which pretrained models can form the VLM that will meet the user's needs most effectively. As shown in Figure 1, no single VLM consistently outperforms the others in accuracy across all dimensions.

VLM capabilities vary significantly depending on their pretrained components Liu et al. (2023c); Xu et al. (2023); Zhang et al. (2024). It is unreliable and unpredictable to rely on human "intuitions" to select pretrained models for the given downstream task, such as selecting the latest one or the most well-known one. Existing model selection methods, such as EMMS Meng et al. (2023) and LogME You et al. (2021), also fall short in the VLM context. These methods are primarily designed for classification, regression, or language-only tasks, where zero-shot performance can serve as an indicator of model quality Brown et al. (2020); Lin et al. (2024); Yi et al. (2024). However, VLMs require vision-text alignment, making zero-shot evaluations unreliable. Without proper alignment, the language model cannot correctly interpret the image embeddings, leading to random and meaningless outputs. With significant constraints on available time and computation cost, it is also unrealistic to try every pretrained model combination and train corresponding VLM candidates. Training a single VLM with vision-text alignment data could take more than 100 GPU hours Liu et al. (2023a); Karamcheti et al. (2024). This motivates the key question of this work: *How to effectively find the best pretrained models in a VLM given a downstream task?*

To address this question, we formulate the *pretrained model selection problem for VLMs* and model it as a resource-constrained task to predict the *alignment* performance of a VLM; i.e., the performance on the downstream task after training the feature projector with the vision-text alignment data. We empirically show that existing model selection methods fail to find the best pretrained model for downstream tasks and a naive approach like grid search is infeasible in practice.

We present **Mordal**, a novel pretrained model selection framework, which automatically and efficiently explores different pretrained model combinations in VLM. Mordal builds on our observation that efficiently solving this problem needs jointly considering two optimization directions: (1) minimizing the number of VLM candidates, where each candidate has different pretrained vision encoders and LLMs; and (2) reducing the evaluation time for each candidate. Overall, we make the following contributions in this work:

- We define the pretrained model selection problem in the context of VLMs and demonstrate that off-the-shelf VLMs often do not contain the best pretrained components for a given downstream task.

- We propose Mordal, an efficient pretrained model search framework, to find the best VLM for a given downstream task. Mordal clusters VLM candidates by their representation similarities while employing early stopping and scaling prediction mechanisms to reduce evaluation time.

- Extensive evaluations show that Mordal efficiently finds the best VLMs with $8.9\times$–$11.6\times$ less computation time than grid search. It also achieves about 69% higher weighted Kendall's $\tau$ on average than the state-of-the-art model selection method across diverse tasks.

## 2 Background and Motivation

We start by outlining the architecture of typical VLMs. Following this, we highlight the challenges in pretrained model selection for VLMs and show that existing VLMs often do not pick ideal pretrained models for downstream tasks. Based on these observations, we present the limitations of potential solutions like grid search, which motivate Mordal's design.

### 2.1 Vision Language Model

Common VLM architectures, like LLaVA Liu et al. (2023a), include a pretrained visual encoder to encode visual features, a pretrained large language model (LLM) to comprehend the user instructions and produce responses, and a vision-language cross-modal feature projector to align the vision encoder outputs to the language models:

**Vision Encoder (VE).**    The vision encoder is responsible for processing input images and extracting relevant features. Potential encoder options are CLIP Radford et al. (2021), SigLIP Zhai et al. (2023), and InternViT Chen et al. (2024b), etc.

Table 1: Evaluation results of capability on tasks of Visual QA, Doc QA, and Knowledge for four VLMs with different pretrained models. CLIP-Vicuna has the same pretrained models and mode structure as LLaVA-1.5-7B Liu et al. (2023a). The best result for each scenario is in **bold** text. There is no silver bullet.

| Model | Vision Encoder | Language Model | Visual QA | | Doc QA | | Knowledge | |
| | | | GQA | VizWiz | ChartQA | DocVQA | ScienceQA | AI2D |
| --- | --- | --- | --- | --- | --- | --- | --- | --- |
| CLIP-Vicuna | CLIP-ViT-L/14 | Vicuna-1.5-7B | 61.5 | 41.2 | 18.2 | **27.6** | 70.4 | 54.8 |
| SigLIP-Vicuna | SigLIP-so400m-patch14 | Vicuna-1.5-7B | **66.4** | **44.8** | **18.4** | 24.1 | 68.5 | 53.0 |
| CLIP-Llama | CLIP-ViT-L/14 | Llama-3-8B | 55.8 | 37.9 | 13.3 | 17.3 | 75.7 | 58.2 |
| SigLIP-Llama | SigLIP-so400m-patch14 | Llama-3-8B | 56.4 | 38.1 | 13.4 | 17.4 | **78.5** | **60.1** |

**Feature Projector (FP).** The vision-language cross-modal feature projector aims to align the encoded image features to the text token embedding space. The feature projector can be achieved directly by a Linear Projector or Multi-Layer Perceptron (MLP) Liu et al. (2023b).

**Large Language Model (LLM).** The language model processes mixed embeddings generated from both user instructions in text as well as image inputs. The commonly used LMs in VLMs are decoder-only LLMs, which include Llama Touvron et al. (2023a), Qwen Yang et al. (2024) and Mistral Jiang et al. (2023).

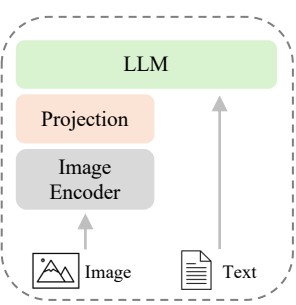

With this structure, a VLM first processes an input image $x_{img} \in \mathbb{R}^{H \times W}$ with an image processor and passes it to a vision encoder $V$. The vision encoder outputs a sequence of raw vision embeddings (or patches) $p_{img} \in \mathbb{R}^{L \times h_{vis}}$ where $p_{img} = V(x_{img})$ and $h_{vis}$ is the hidden state dimension of the vision encoder outputs. The feature projector $P$ will then map $p_{img}$ to aligned vision embeddings $e_{img} \in \mathbb{R}^{L \times h_{text}}$ where $h_{text}$ is the hidden dimension of the corresponding LM token. The aligned embeddings $e_{img}$ will append to text prompt embedding $e_{text} = embed(prompt)$ and feed into the LLM to generate output text.

Figure 2: Overview of VLM architecture.

## 2.2 Pretrained Model Selection: No Silver Bullet

VLMs are versatile and powerful because most vision tasks can be formulated as next-token prediction. To train a VLM for a specific downstream task, developers usually *cherry-pick* pretrained vision encoders and language models for alignment. However, different pretrained models have varying capacities, which affect VLM performance. To investigate the impact of different pretrained models, we conduct grid search on 49 VLM candidates (i.e., seven vision encoders and seven language models) and train each candidate with *the same alignment data*. While complete evaluation is described in Section 4, we present the performance of four representative VLM candidates on six datasets, across three dimensions: Visual QA (GQA Hudson & Manning (2019) and VizWiz Gurari et al. (2018)), Doc QA (ChartQA Masry et al. (2022) and DocVQA Mathew et al. (2021)) and Knowledge (ScienceQA Lu et al. (2022) and AI2D Kembhavi et al. (2016)). As shown in Table 1, both pretrained vision encoders and language models have a significant impact on VLM performance. We further show in Appendix A that fine-tuning on specific datasets does not eliminate these differences, since model performance remains bounded by the quality of its pretrained components. Overall, *there is no silver bullet pretrained model or model combination that reigns supreme for all tasks*.

Given these observations, one may naturally ask: *how can we select the best combination of pretrained models for a specific task?* In this paper, we address the pretrained model selection problem in the context of VLMs. Given an alignment dataset and a target task, we aim to find the combination of a pretrained vision encoder and a language model that achieves the best performance on the target task after alignment.

## 2.3 Limitations of Existing Solutions

Solving the pretrained model selection problem by relying on empirical experiences or intuitions, such as choosing the newest, largest or most well-known model, is unreliable. For example, in Table 1, a VLM with Vicuna-1.5-7B can outperform a VLM with LLaMA-3-8B in certain settings.

Existing model selection methods, such as EMMS Meng et al. (2023), LogME You et al. (2021), LEEP Nguyen et al. (2020a), and NLEEP Li et al. (2021), primarily focus on tasks involving classification or regression Brown et al. (2020). These methods typically assess models based on transferability metrics, which are inadequate for VLMs requiring alignment Vu et al. (2020); Yi et al. (2024); Lin et al. (2024). VLMs like LLaVA Liu et al. (2023a), as shown in Figure 2, involve combining vision encoders with large language models (LLMs) through a feature projector to achieve multimodal interaction. Without alignment, the LLM cannot interpret image embeddings, rendering transferability metrics less effective.

Performing an exhaustive search for pretrained model selection is also computationally prohibitive. Even within a manageable search space, training a single VLM candidate with the LLaVA-1.5 dataset, using a pretrained 7B LLM backbone, can consume over 100 GPU hours. Considering the vast number of pretrained models available (e.g., over 150,000 LLMs on HuggingFace as of January 2025), exhaustively evaluating every candidate becomes impractical. The cost is further exacerbated as new encoders and LLMs continue to be introduced, necessitating constant re-evaluation to integrate new components.

To address the inefficiencies of exhaustive search in pretrained model selection, we must reduce the search cost along two key dimensions: (1) reducing the number of candidates and (2) minimizing the time required to evaluate each candidate. By optimizing these two dimensions, the exhaustive search process can be replaced with a more efficient pipeline that balances time consumption and selection accuracy.

## 3  Mordal Design

This section details the core components of Mordal's design. The exhaustive search is expensive because it (1) needs to evaluate every candidate (large search space), and (2) needs to train each candidate with a full dataset to see its performance (high evaluation cost). Mordal reduces search space by clustering the candidates based on their similarity and by introducing a two-step inter- and intra-cluster evaluation (§3.1), and reduces evaluation cost of each candidate with early stopping and scaling prediction (§3.2).

### 3.1  Candidate Clustering

With the rapid increase in the number of pretrained models in popular model zoos (e.g., Hugging-Face Face (2025)), evaluating every candidate combination is expensive. Based on prior observations that similar models tend to have similar performance Hu et al. (2023); Yu et al. (2024); Lai et al. (2023), Mordal clusters candidates and evaluates them in two steps: inter-cluster and intra-cluster, to reduce the search space.

**Measuring similarity.**  Measuring the similarity of VLM candidates – without training projectors between vision encoders and language models – is challenging. Parameter similarity Lai et al. (2023), which has been used to measure similarity between model architectures, does not fully consider the data distribution pattern in the target task. Models with high parameter similarity could still show different performance on different tasks. Therefore, in Mordal, we consider *representation similarity* between VLM candidates, which depends on the target task.

Mordal employs centered kernel alignment (CKA) Kornblith et al. (2019) to evaluate the similarity of representations between two VLM model structures. CKA has been proven to be an effective tool for understanding and comparing the information encoded across different layers of neural networks. Formally, CKA operates on two datasets by analyzing their corresponding activation matrices. The CKA score is defined as:

$$\text{CKA}(K, L) = \frac{\text{HSIC}(K, L)}{\sqrt{\text{HSIC}(K, K) \cdot \text{HSIC}(L, L)}} \tag{1}$$

where K and L are the kernel matrices of activations of two models. HSIC is the Hilbert-Schmidt Independence Criterion (HSIC) defined as:

$$\text{HSIC}(K, L) = \text{Tr}(KHLH) \tag{2}$$

where $\text{Tr}()$ is the trace of a matrix and H is the centering matrix $H = I - \frac{1}{n}\mathbf{1}\mathbf{1}^\top$. CKA is particularly useful in this context for two reasons. First, CKA can compare representations with differing shapes

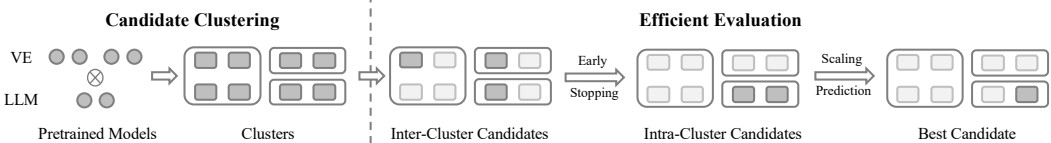

Figure 3: An overview figure for Mordal. Gray shapes represent pretrained models and VLM candidates, while white blocks are eliminated ones. Mordal clusters similar candidates, evaluates one per cluster, eliminates weak clusters, and uses regression to predict the best candidate.

| Vision Encoder | ScienceQA | VizWiz |
|---|---|---|
| CLIP-ViT-L/14 | 67.6 | 41.2 |
| SigLIP-so400m-patch14 | 67.7 | 44.8 |
| DFN5B-CLIP-ViT-H/14 | 62.3 | 34.4 |
| InternViT-300M | 56.9 | 30.7 |

(a) Evaluation Results       (b) ScienceQA    (c) VizWiz

Figure 4: Evaluation results of VLMs with different vision encoders and the same language model (i.e., Vicuna-1.5-7B). Similarity scores between four vision encoders on ScienceQA and VizWiz.

generated by different pretrained models, a task where traditional metrics such as cosine similarity fail. Second, as vision representations are commonly projected through MLP layers, this transformation does not compromise CKA's properties, making it robust and well-suited for such evaluations.

To validate the effectiveness of CKA, we train multiple VLM combinations with four vision encoders (CLIP [43], SigLIP [58], DFN-CLIP [12], InternViT [7]) and the same LLM backend Vicuna-1.5-7B. The trained VLMs are evaluated on two different datasets: ScienceQA and VizWiz. As shown in Figure 4, the image representation (i.e., outputs of projection heads) generated by different vision encoders show different levels of similarity to each other. For example, with input images from ScienceQA, CLIP, SigLIP, and DFN-CLIP generate similar representations. Meanwhile, DFN, CLIP and InternViT generate similar representations in VizWiz. From the results in Figure 4a, the vision encoders with similar representations will have similar performance with the same language model. Although this observation does not allow us to directly predict the target task's performance, it helps reduce the number of VLM candidates by eliminating similar pretrained models.

**Two-step clustering.** Computing the CKA score between each pair of candidates can be expensive, since $K$ and $L$ are activations from batched data input. To reduce the amount of pair-wise CKA computation, Mordal introduces a two-step VLM clustering strategy: (1) clustering vision encoders, (2) clustering language models based on a fixed vision representation. We detail the clustering process as follows (see full pseudocode in Appendix B):

- *Vision encoder clustering.* Mordal computes the representation similarity between vision encoders using CKA. A distance matrix $Dist_{ve}$ is then constructed based on the dissimilarity values. The clustering function will take an input threshold $t_{ve}$ and output the vision encoder clusters $\mathcal{C}_{ve}$.

- *Language model clustering.* Language model clusters are constructed based on vision representations of each vision encoder cluster. Using the medoid vision encoder from each cluster, Mordal generates a fixed image embedding for the dataset and a warmed-up feature projector transforms the shape of image embeddings to match the LLM input shape. A distance matrix $Dist_{llm}$ will record the dissimilarity and language models are then clustered based on an input threshold $t_{llm}$.

For each vision encoder cluster and the corresponding language model clusters, Mordal generates the candidate clusters by conducting the Cartesian product of the two clusters. The two-step clustering process reduces the computation costs by avoiding computing the similarity between candidates that have dissimilar vision encoders, which we show in Section 4.3 that usually yield distinct performance.

**Inter- and intra-cluster evaluation.** After grouping the candidates into clusters, Mordal finds the best candidate with two-step evaluation: inter-cluster evaluation and intra-cluster evaluation.

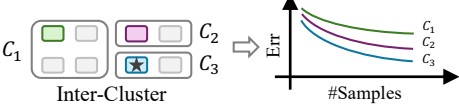 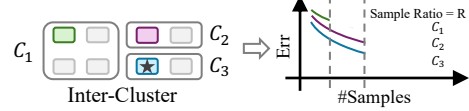

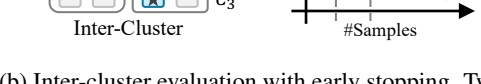

(a) Inter-cluster evaluation: selects top-$K$ clusters. When $K = 1$, $C_3$ is selected.

(b) Inter-cluster evaluation with early stopping. Two candidates are early stopped and $C_3$ is selected.

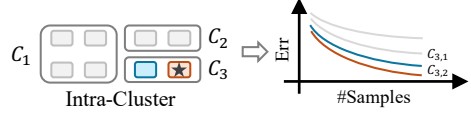 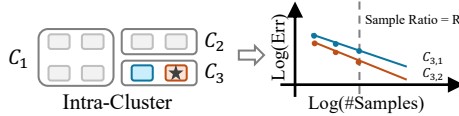

(c) Intra-cluster evaluation: identifies the best candidate. Among the remaining two candidates, $C_{3,2}$ is selected.

(d) Intra-cluster evaluation with scaling prediction. $C_{3,2}$ is selected based on the predicted performance.

Figure 5: Inter- and intra-cluster evaluation over candidate clusters. We accelerate selection by applying early stopping during inter-cluster evaluation and scaling prediction during intra-cluster evaluation. Colors denote different VLM candidates, and the star marks the selected best-performing candidate.

The detailed process is shown in Figures 5a and 5b. Given that candidates in the same cluster have similar performance, inter-cluster evaluation first picks medoid from each cluster as the *representative candidate* and compares *performance of each cluster* to eliminate poorly performing clusters. The remaining Top-$K$ clusters will be evaluated in the second step, where $K$ is a user-defined parameter.

With the inter-cluster evaluation, many fewer candidates remain in consideration. Intra-cluster evaluation goes back to candidate-granularity evaluation by aggregating candidates from the remaining Top-$K$ clusters. It trains all of them on the given alignment dataset, and returns the one with the best performance to the user.

### 3.2 Efficient Evaluation

Section 3.1 reduces the search space; however, evaluating each candidate in both stages (i.e., inter- and intra-) still requires training each candidate with a full dataset, which remains expensive. To address this, Mordal integrates two complementary techniques: early stopping for inter-cluster pruning, and scaling prediction for efficient intra-cluster evaluation. These components are co-designed to reuse intermediate checkpoints, further reducing redundant computation.

**Early stopping.** Mordal adopts the Successive Halving Algorithm (SHA) Jamieson & Talwalkar (2016) to aggressively eliminate low-quality candidates during inter-cluster evaluation. As shown in Figure 5c, SHA is conducted during inter-cluster evaluation, where all representative candidates are evaluated with the maximum data sample ratio $R$. It consists of multiple rounds, also known as *rung*. In each round, Mordal allocates a budget $b$ to each candidate, evaluates all of them, and keeps the top $1/\eta$ candidates. In the next round, Mordal increases the budget to $b \times \eta$ per candidate. This repeats until representative candidates are converged or Top-$K$ candidates are determined. In cases where the number of remaining candidates is large, Mordal also applies SHA again to intra-cluster evaluation. With SHA as the early stopping mechanism, Mordal *eliminates* poor-performing candidates earlier. This mechanism concentrates resources on the most promising clusters and produces intermediate checkpoints that can be reused downstream.

**Scaling prediction.** Scaling laws have traditionally been used to characterize the relationship between model size, training data, and performance in language models Kaplan et al. (2020); Hoffmann et al. (2022). These laws are typically formulated as a power-law relationship between LLMs' cross-entropy loss $L$ and their compute scale measures, which take the form:

$$L(N, D) = \frac{a}{N^\alpha} + \frac{b}{D^\beta} + e \tag{3}$$

where $N$ the number of model parameters, and $D$ the number of training samples. The fitted parameters $(\alpha, \beta, a, b, e)$ are derived from training models across different scales. These relationships

are typically log-linear under a log-log transformation and have been exploited for model selection in LLMs Lin et al. (2024); Ruan et al. (2024). In contrast to prior work, our focus is on alignment performance in VLMs, where the model size $N$ is fixed, but the alignment data size $D$ is varied. We find that a similar log-linear trend exists between the size of alignment data and task-specific error metrics (details are provided in Section 4.3). This observational scaling law enables us to estimate a candidate VLM's full-data performance from a small number of low-cost training runs.

With the observational scaling law, Mordal employs a scaling prediction algorithm that automatically detects the log-linear scaling with sampled alignment dataset to conserve resources. As shown in Algorithm 1, for each candidate $c$ in remaining candidate list $C$, the algorithm starts from a maximum data sample ratio $R$ (e.g., $\frac{1}{8}$). It evaluates a checkpoint trained on randomly sampled data with ratio $R$. The corresponding performance point $(Log(r), Log(Err))$ will be recorded in list $P$. After that, the algorithm reduces data sample ratio and repeats the above process until enough performance points (i.e., $p$) are collected and the log-linear relationship is observed for a given candidate $c$. Since data sample ratio $r$ is decreasing, we may effectively start the evaluation of $r/u$ from existing intermediate checkpoints to save computation costs.

---

**Algorithm 1:** Scaling Prediction

1 **Input**    :Maximum data sample ratio $R$, scaling ratio $u$, minimum required point $p$, VLM candidate $c$
2 **Output** :Prediction result
3 Initialize loss-size pair list $P = []$
4 Initialize data sample ratio $r = R$
5 **while** *True* **do**
6     /* Evaluate performance*/
7     $Err = Evaluate(\mathcal{D}_{align}, \mathcal{D}_{task}, c, r)$
8     $P.append((log(r), log(Err)))$
9     **if** $|P| > p$ **then**
10       Fit a linear regression model $f_c$ on $P$
11       Break if $f_c$ fitting loss $> \delta$
12     **end**
13     $r = r/u$ /* Reduce data samples*/
14 **end**
15 Return $f_c(1)$

---

Scaling prediction complements Mordal's two-stage evaluation and helps reduce search cost. By fitting a log-linear model $f_c$ on a few sampled alignment points $P$, Mordal can estimate the full-data performance $f_c(1)$ without fully training the candidate $c$. This enables early elimination of weaker candidates while prioritizing promising ones. Since the evaluation proceeds from larger to smaller sample ratios, intermediate checkpoints can be reused to save additional GPU time.

# 4 Evaluation

We conducted extensive experiments to thoroughly evaluate Mordal's performance. Our implementation builds on Cornstarch Jang et al. (2025), a multimodal training framework that includes distributed training features. These experiments assessed Mordal's effectiveness in pretrained model selection and included an ablation study. The key findings are summarized as follows:

1. Mordal identifies the best combination of vision encoder and LLM for the target task $8.9\times$-$11.6\times$ faster than the grid search. It also achieves about 69% higher weighted Kendall's $\tau$ on average than the state-of-the-art model selection method across diverse tasks.

2. We further validate the effectiveness of observational scaling law. By conducting the ablation studies, we show that each component in Mordal helps reduce the total search time while ensuring that it can still identify the top-performing candidates.

## 4.1 Experimental Setup

The experiments are conducted on seven mainstream datasets across three domains with seven vision encoders and seven LLMs. We deployed Mordal on a set of VMs on a cluster with a total of 16 NVIDIA A40 GPUs. Each GPU has 48 GB GDDR6 memory.

**Dataset.** We use LLaVA-1.5-Instruction dataset Liu et al. (2023a) for alignment. In practice, users may use their own alignment datasets. For target evaluation, we first select six standard benchmarks across Visual QA, Doc QA, and Knowledge tasks. To test broader generality, we further include MMMU, a multi-disciplinary benchmark covering diverse real-world domains. These datasets are commonly used to assess VLM performance Zhang et al. (2024).

**Model zoo and training settings.** We select seven vision encoders and seven language models based on popularity and performance. The selected vision models include both *language-supervised*

Table 2: Summary of improvements. Search Time (h) and Top-1 Model Quality (%) results for different datasets. For each task, grid search exhaustively evaluates 49 candidates, requiring *5439 GPU hours*. Mordal significantly reduces the amount of training needed (i.e., GPU Saving) while successfully finding the Top-1 VLM candidate.

| Task | Dataset | CLIP-Vicuna (LLaVA-1.5) | Grid Search | | Mordal (ours) | | |
|------|---------|------------------------|-------------|------------|---------------|---------|-------------|
| | | | Top-1 Score | Model Name | Time | Speedup | Top-1 Score |
| Visual QA | GQA Hudson & Manning (2019) | 61.5 | 66.4 | SigLIP-Vicuna | 483 | 11.2× | **66.4** |
| | VizWiz Gurari et al. (2018) | 41.2 | 46.9 | SigLIP-Mistral | 469 | 11.6× | **46.9** |
| Doc QA | ChartQA Masry et al. (2022) | 18.2 | 20.2 | CLIP-Qwen | 607 | 8.9× | 18.6 |
| | DocVQA Mathew et al. (2021) | 27.6 | 28.5 | SigLIP-Qwen | 593 | 9.2× | **28.5** |
| Knowledge | ScienceQA Lu et al. (2022) | 70.4 | 78.5 | SigLIP-Llama | 472 | 11.5× | **78.5** |
| | AI2D Kembhavi et al. (2016) | 54.8 | 65.2 | SigLIP-Qwen | 496 | 10.9× | **65.2** |
| Multi-discipline | MMMU Yue et al. (2024) | 35.3 | 36.6 | SigLIP-Llama | 503 | 10.8× | **36.6** |

(a) Grid Search (GQA)  (b) Mordal (GQA)  (c) Grid Search (AI2D)  (d) Mordal (AI2D)

Figure 6: Top-5 candidates from grid search and Mordal on GQA and AI2D.

*models* (e.g., OpenAI CLIP) and *self-supervised models* (e.g., DINOv2). For LLMs, all models follow the decoder-only structure, and we pick the most commonly used 7B LLMs from *HuggingFace*. We follow the same structure of LLaVA-1.5-7B, which has a two-layer MLP projector, and finetune pretrained LLM with LoRA. The default training setting uses Adam optimizer with minibatch size 4 and initial learning rate 1e-4. We use the linear schedule to decrease the learning rate linearly from the initial value. The complete list of models and additional training details are provided in Appendix D.

**Baseline.** We use grid search as the primary baseline, where every VLM candidate is fully trained and evaluated to find the top-1 model. We also measure the total training time, in GPU hours, required by both grid search and Mordal. We further compare Mordal with four model selection baselines: EMMS Meng et al. (2023), LogME You et al. (2021), LEEP Nguyen et al. (2020a), and NLEEP Li et al. (2021). For a fair comparison, we use the same alignment data and ensure that the total training time allocated to each candidate matches Mordal's overall budget. This allows us to assess both accuracy and efficiency under equal resource constraints. Implementation details can be found in Appendix D.

**Metric.** To evaluate ranking accuracy, we compare candidates ranked by performance in grid search with those ranked by elimination order in Mordal. Ideally, if a candidate ranks higher in grid search, it should also rank higher in Mordal. This can be captured by Kendall's $\tau$ coefficient defined as:

$$\tau = \frac{2}{M(M-1)} \sum_{1 \leq i < j \leq M} sgn(T_i - T_j) sgn(S_i - S_j) \tag{4}$$

where $M$ is the total number of candidates and $sgn()$ is the sign function. A perfect ranking match results in $\tau = 1$. To further focus on top-performing candidates, we adopt the weighted Kendall's coefficient $\tau_w$, which is previously used in You et al. (2021); Vigna (2015). The details for implementing $\tau_w$ can be found in SciPy Virtanen et al. (2020).

## 4.2 Performance Results

This sections evaluates Mordal's effectiveness in identifying top-performing VLM candidates under constrained training budgets. We also report the weighted Kendall's $\tau$ between Mordal's candidate rankings and baselines' candidate rankings to assess ranking quality.

**Mordal is significantly faster than grid search.** Table 2 shows that grid search requires 5439 GPU hours to exhaustively train 49 VLM candidates per task. In contrast, Mordal reduces the total search time by 8.9× to 11.6× across tasks, with search time varying by cluster formation and candidate

Table 3: Comparison of different model selection methods with the same time consumption. Kendall $\tau$ represents the differences of top-performing candidates compared with the groundtruth (i.e., grid search) and *larger $\tau$ is better*.

| Task | Dataset | EMMS | LogME | LEEP | NLEEP | Mordal (Ours) |
|------|---------|------|-------|------|-------|---------------|
| Visual QA | GQA Hudson & Manning (2019) | 0.682 | -0.162 | 0.232 | 0.435 | **0.814** |
| | VizWiz Gurari et al. (2018) | 0.657 | 0.236 | 0.351 | 0.502 | **0.882** |
| Doc QA | ChartQA Masry et al. (2022) | 0.238 | -0.144 | -0.071 | 0.298 | **0.765** |
| | DocVQA Mathew et al. (2021) | 0.172 | 0.155 | 0.111 | 0.265 | **0.897** |
| Knowledge | ScienceQA Lu et al. (2022) | 0.770 | 0.269 | 0.344 | 0.562 | **0.960** |
| | AI2D Kembhavi et al. (2016) | 0.557 | 0.193 | 0.316 | 0.614 | **0.894** |
| Multi-discipline | MMMU Yue et al. (2024) | 0.526 | 0.101 | 0.245 | 0.220 | **0.875** |

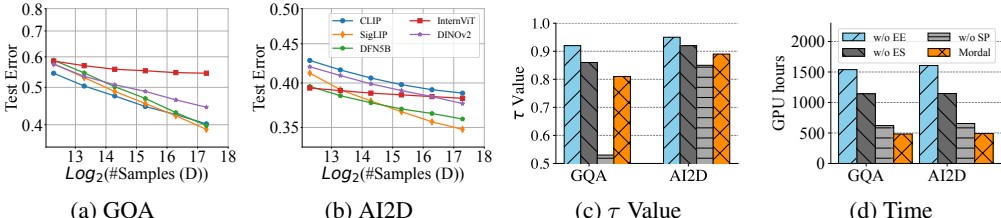

(a) GQA      (b) AI2D      (c) $\tau$ Value      (d) Time

Figure 7: Observational scaling law validation and ablation study results for efficient evaluation (EE), early stopping (ES) and scaling prediction (SP). The results are on GQA and AI2D.

convergence rates. Despite this reduction, Mordal successfully identifies the top-1 model in six out of seven tasks. For example, on VizWiz, Mordal completes the search in 469 GPU hours and selects SigLIP-Mistral, which achieves 46.9% accuracy. Notably, for most tasks, the best VLM candidates have different pretrained model combinations and all of them surpass the performance of the default LLaVA-1.5-7B structure, highlighting the importance of pretrained model selection.

**Mordal outperforms existing model selecing methods in finding top-performing models.** Mordal consistently outperforms existing model selection methods (EMMS, LogME, LEEP, and NLEEP) by achieving higher weighted Kendall's $\tau$ scores across all seven datasets, as shown in Table 3. For example, on the ScienceQA dataset, Mordal achieves a $\tau$ value of 0.96, significantly surpassing the best baseline (EMMS) with a $\tau$ of 0.77. This is because Mordal effectively captures the alignment performance of VLM combinations, while baseline methods rely on features from foundation models that do not fully distinguish between different model pairings. However, Mordal can occasionally miss the best combinations when promising candidates are grouped with weaker ones or are prematurely excluded by the early stopping mechanism (See Figure 6). Despite this limitation, Mordal consistently identifies top-performing models more accurately and efficiently than traditional methods.

### 4.3 Ablation Studies

In this section, we first validate observational scaling law in Mordal. Then we conduct a comprehensive ablation study to evaluate the candidate clustering and efficient exploration in Mordal and their contributions to overall performance.

**Observation scaling law validation.** To validate the existence of scaling laws in VLM alignment, we train multiple VLM combinations with different vision encoders and the same language model Qwen2-7B on the sampled LLaVA-1.5-Instruction dataset. The trained VLMs are evaluated on two different datasets: GQA and AI2D. As shown in Figures 7a and 7b, we observe log-linear scaling across different VLMs, which supports the design of scaling prediction. However, the log-linear scaling will only appear after a certain number of training samples, which is consistent with the conclusion in previous work Lin et al. (2024); Ruan et al. (2024).

**Effect of candidate clustering.** Candidate clustering plays a vital role in Mordal as it enables inter- and intra-cluster evaluation. As illustrated in Figure 7d, inter- and intra-cluster evaluation (i.e., Mordal without efficient evaluation) significantly reduces training time while maintaining a

high $\tau$ value. By grouping candidates with similar characteristics into clusters, Mordal evaluates representative candidates from each cluster first and eliminates candidates in poor-performed clusters.

**Effect of efficient exploration.**    Early stopping mechanism prunes candidates during the early stage of training. While it significantly reduces the search time, applying it during the entire evaluation (i.e., Mordal without scaling prediction) will eliminate some promising candidates (e.g., SigLIP-Qwen on AI2D shown in Figure 7b). It evaluates candidates based on intermediate performance and leads to a low $\tau$ value. Mordal limits the usage of early stopping and introduces scaling prediction instead to predict the performance of promising candidates. As shown in Figure 7c, this leads to a significant improvement in the $\tau$ value while further reducing the total training time.

### 4.4 Sensitivity Analysis

Sensitivity analysis explores how key hyperparameters affect Mordal's performance and efficiency, focusing on clustering thresholds $t_{ve}$ and $t_{llm}$, exploration parameters $s_{inter}$ and $s_{intra}$, and scaling prediction $p$. Careful tuning of these parameters ensures efficient operation without compromising Mordal's ability to select top-performing models. Generally, Mordal is robust and consistently identifies the best-performing model across most hyperparameter settings.

| Category | Config | Time | Top-1 Score | $\tau$ |
|---|---|---|---|---|
| Mordal | Default | 483 | 66.4 | 0.81 |
| Candidate | $t_{ve} = 0.5$ | 446 | 66.4 | 0.52 |
| Clustering | $t_{ve} = 0.9$ | 1041 | 66.4 | **0.86** |
| Inter-Cluster | $topk_{inter} = 2$ | 417 | 66.4 | 0.73 |
| Evaluation | $topk_{inter} = 4$ | 564 | 66.4 | 0.83 |
| Intra-Cluster | $topk_{intra} = 2$ | **451** | **66.4** | 0.81 |
| Evaluation | $topk_{intra} = 4$ | 522 | 66.4 | 0.81 |
| Prediction | $p = 4$ | 501 | 66.4 | 0.81 |

Table 4: Summary of sensitivity analysis for GQA.

**Effect of clustering hyperparameters.**    The clustering threshold $t_{ve}$ and $t_{llm}$ significantly affects Mordal's performance and efficiency. As shown in Table 4, a smaller threshold $t_{ve} = 0.5$ creates fewer, larger clusters based on the LLM's general characteristics, which will lower the $\tau$ value by missing finer distinctions and discarding strong candidates with other LLM backend. On the other hand, a larger threshold $t_{ve} = 0.9$ results in more, smaller clusters, capturing subtle differences and improving the $\tau$ value but increasing the number of candidates to evaluate during scaling prediction, leading to longer search time. Balancing the threshold is crucial to ensure diverse clusters while keeping the computational cost reasonable.

**Effect of inter- and intra-cluster hyperparameters.**    Based on Table 4, inter-cluster exploration generally has a greater impact on Mordal's performance than intra-cluster exploration. A smaller $topk_{inter}$ reduces the number of clusters evaluated, speeding up the search but lowering the $\tau$ value by excluding promising clusters aggressively. A larger $topk_{inter}$ explores more clusters, increasing search time but improving the $\tau$ value by retaining diverse clusters. Intra-cluster early stopping affects candidate selection within clusters, with smaller $topk_{intra}$ focusing on fewer candidates and larger $topk_{intra}$ exploring more candidates for scaling prediction. However, its influence is smaller, as clusters already limit diversity. Properly balancing these $topk_{inter}$ and $topk_{intra}$ ensures efficient exploration and strong performance.

**Effect of scaling prediction hyperparameters.**    The scaling prediction parameter $p$ has minimal impact on Mordal's overall performance but affects search time. Increasing $p$ beyond appropriate values adds to the computational cost without improving results. In practice, $p = 3$ is sufficient for constructing the linear regression model used in scaling prediction.

## 5   Conclusion

We presented Mordal, an automated framework for efficient pretrained model selection in vision-language models (VLMs). Mordal reduces search cost by pruning candidate combinations and minimizing evaluation time per model. Experiments show that Mordal achieves up to $11.6\times$ speedup over grid search while maintaining top-1 accuracy, and outperforms existing baselines in ranking quality. This highlights the effectiveness of task-aware, alignment-required model selection for VLM deployment.

## Ethics Statement

This work adheres to the ICLR Code of Ethics. Our study does not involve human subjects or sensitive personal data. The datasets we use are publicly available and widely adopted in the research community. We do not anticipate any harmful societal impact from our methodology, and we believe that the contributions of this work will positively benefit the community.

## Reproducibility Statement

We have taken steps to ensure reproducibility of our results. Detailed descriptions of datasets, experimental settings, and evaluation protocols are included in the main text and appendix. Model configurations, hyperparameters, and training details are provided. In addition, we will release the source code, model checkpoints, and full evaluation results in the future. We believe this will enable other researchers to fully reproduce and extend our work.

## Acknowledgements

We thank the ICLR reviewers, as well as members of SymbioticLab, for their helpful feedback. This work was supported in part by NSF grants CCF-2450085 and CNS-2106184, and by grants from Ford and Cisco.

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

# A    Task-Specific Fine-Tuning Does Not Eliminate Performance Differences

Table 5: Task-specific fine-tuning results across three datasets.

| Model | GQA | ChartQA | ScienceQA |
|---|---|---|---|
| CLIP-Vicuna | 62.8 | 24.4 | 71.3 |
| SigLIP-Vicuna | **67.4** | **26.3** | 70.2 |
| CLIP-LLaMA | 56.5 | 22.9 | 78.1 |
| SigLIP-LLaMA | 58.7 | 23.2 | **81.5** |

An important question is whether performance differences across VLMs disappear after task-specific fine-tuning. Prior studies in both single-modality and multimodal settings indicate that this is not the case: performance differences often persist even after fine-tuning. For example, Lin et al. (2024) and Zeng et al. (2025) show that not all pretrained LLMs converge to similar performance after fine-tuning, even with identical data and compute budgets. Our work extends this investigation to the multimodal domain.

We conducted 12 task-specific fine-tuning runs: the four representative VLM combinations from Table 1 of the paper were fine-tuned on three target datasets that differ in their overlap with the LLaVA-1.5 alignment mixture—GQA (in-distribution), ChartQA, and ScienceQA (out-of-distribution). Results are shown in Table 5. Even after fine-tuning on data from each target distribution, we still observe performance differences of up to 10–11 percentage points among models. Notably, the top-1 model remains consistent with the selection reported in Table 1 of the paper. These findings indicate that Mordal generalizes well with the alignment data mixture and remains effective under distribution shift. Practically, this result highlights that even if a model is fine-tuned, its performance is bounded by the quality of its pretrained components. Thus, selecting strong pretrained VLMs remains crucial. When sufficient fine-tuning data is available, Mordal can also be applied directly on the target data to prune candidates, further reducing model selection time and compute.

# B    Algorithm

---

**Algorithm 2:** Candidate Clustering

**Input** : Target task $D$, Model Zoo $\mathcal{M}_{ve}$ and $\mathcal{M}_{llm}$, clustering threshold $t_{ve}$ and $t_{llm}$
**Output** : Candidate clusters $C_{vlm}$

1 **Def** *CandidateClustering($\mathcal{M}_{ve}, \mathcal{M}_{llm}$)*:
2     /* Vision Encoder Clustering */
3     **for** $M_A, M_B \in \mathcal{M}_{ve}$ **do**
4        $dist_{A,B} = 1 - CKA(\mathcal{D}, M_A, M_B)$
5        $Dist_{ve}[M_A][M_B] = Dist_{ve}[M_B][M_A] = dist_{A,B}$
6     **end**
7     $\mathcal{C}_{ve} = Clustering(Dist_{ve}, t_{ve})$
8     /* Candidate Clustering */
9     **for** $C_A \in \mathcal{C}_{ve}$ **do**
10        $M_{medoid} = PickMedoidModel(C_A)$
11        /* LLM Clustering */
12        **for** $M_A, M_B \in \mathcal{M}_{llm}$ **do**
13           $dist_{A,B} = 1 - CKA(M_{medoid}(\mathcal{D}), M_A, M_B)$
14           $Dist_{llm}[M_A][M_B] = Dist_{llm}[M_B][M_A] = dist_{A,B}$
15        **end**
16        $\mathcal{C}_{llm} = Clustering(Dist_{llm}, t_{llm})$
17        $\mathcal{C}_{vlm}.append(C_A \times \mathcal{C}_{llm})$
18     **end**
19     Return $\mathcal{C}_{vlm}$

---

Detailed two-step clustering algorithm for candidate clustering as described in Section 3.1. We adopt the MinibatchCKA for computation efficiency, which is introduced in Nguyen et al. (2020b) and later used in Raghu et al. (2021). In LLM clustering, we use the last hidden state from LLM as the sentence representation for CKA computation since it produces the best clustering performance. We

leverage the hierarchical clustering from `scipy.cluster.hierarchy` library in SciPy Virtanen et al. (2020). It is possible to adopt other clustering methods. Overall, this approach significantly reduces the number of candidates to explore.

## C  Implementation

```python
import mordal

def search_with_mordal(model_zoo, alignment_data, target_task_data):
    model = mordal.query_for_model(
    data=alignment_data, # LLaVA-1.5-Mixture
    task=target_task_data, # GQA
            pretrained_ve_zoo=model_zoo['ve'],
            pretrained_llm_zoo=model_zoo['llm'],
    vlm_kwargs={
    'projector': 'MLP', 'freeze_ve': True, 'freeze_llm': False,
            },
    mordal_kwargs={
    'clustering': {'t_ve': 0.7, 't_llm': 0.8},
    'exploration': {'top_k_inter': 3, 'top_k_intra': 3},
    'early_stopping': {'R': 0.125, 'b': 0.03, 'eta': 2}
    'scaling_prediction': {'R': 0.125, 'u': 2, 'delta': 0.01}
            }
    )
```

Listing 1: Mordal interface.

This section describes Mordal's implementation details and configurations to ensure efficient and scalable pretrained model selection. We implement Mordal's candidate training pipeline on top of Cornstarch Jang et al. (2025), a multimodal training framework that enables efficient distributed execution. We highlight key design choices that optimize resource utilization without compromising performance. As shown in code snippet Listing 1, to submit a job, users need to provide the alignment data and data for the target task. Users are also allowed to submit a list of available pretrained models. In `vlm_kwargs`, users may specify the projector's architecture and whether to free pretrained components. When unfreezing pretrained components, instead of performing expensive full finetuning, we uses Low-Rank Adaptation (LoRA) Hu et al. (2021) implemented by Parameter-Efficient Fine-Tuning (PEFT) Mangrulkar et al. (2022) and manage LoRA configurations for each pretrained model. LoRA injects task-specific adaptations into the pretrained model by learning low-rank updates for certain layers while keeping the core parameters frozen. This significantly reduces computational and memory overhead, making finetuning feasible under resource-constrained settings. We incorporate Flash Attention Dao (2023) for scalable attention computation, which is a memory-efficient implementation of scaled dot-product attention that avoids redundant operations and reduces memory overhead. All models are trained with torch bfloat16 precision, which balances computational efficiency and numerical stability. Mordal also automatically allocates idle GPU resources to candidates that are not converged to speed up the exploration process.

## D  Additional Experiments

| Vision Encoders | LLMs |
| --- | --- |
| CLIP-ViT-L/14@336 Radford et al. (2021) | Vicuna-1.5-7B Chiang et al. (2023) |
| SigLIP-so400m-patch14@384 Zhai et al. (2023) | Llama-2-7B Touvron et al. (2023b) |
| DFN-CLIP-ViT-H/14@378 Fang et al. (2023) | Llama-3-8B Dubey et al. (2024) |
| InternViT-300M/14@448 Chen et al. (2024b) | Mistral-v0.2-7B Jiang et al. (2023) |
| DINOv2-ViT-L/14@518 Oquab et al. (2023) | Qwen2-7B Yang et al. (2024) |
| EVA-CLIP-02-ViT-L/14@336 Sun et al. (2023) | Phi-3-Small-7B Abdin et al. (2024) |
| ConvNeXt-L/14@256 Ilharco et al. (2021) | Gemma-1.1-7B Team et al. (2024) |

Table 6: List of vision encoders and LLMs in experiments.

**Model zoo.** We evaluate seven vision encoders and seven language models as shown in Table 6. Most models are available on *HuggingFace* Face (2025) while EVA-CLIP and ConvNeXt are supported by *timm* library Wightman (2019). For ConvNeXt, we interpolate the output embeddings to 16x16 patches following Cambrian-1 Tong et al. (2024).

**Hyperparameter settings.** For pretrained model clustering, the threshold for vision encoder and LLMs are set to $t_{ve} = 0.7$ and $t_{llm} = 0.8$, respectively. And the warmup round for the feature projector is 10. When performing an efficient evaluation, we set both $topk_{inter}$ and $topk_{intra}$ to 3, which means that the Top-3 clusters will be selected in inter-cluster evaluation and Top-3 candidates will be selected in intra-cluster evaluation with the early stopping mechanism. We use $\eta = 2$ as the default reduction factor, which is consistent with typical SHA settings. We further set $p = 3$ and $\delta = 5e - 5$ by default for scaling prediction. We discuss the effect of hyperparameters in Section 4.4.

**Baseline.** We compare Mordal with four model selection baselines: EMMS Meng et al. (2023), LogME You et al. (2021), LEEP Nguyen et al. (2020a), and NLEEP Li et al. (2021). For a fair comparison, we use the same alignment dataset and ensure that the total training time allocated to all candidates matches Mordal's overall GPU hour budget. Specifically, we divide Mordal's total training time by the number of candidates, and assign this budget to each candidate. Each VLM candidate is partially trained using this fixed alignment time. We then extract features from the trained model and compute label representations using three foundation models: CLIP Radford et al. (2021), BERT Devlin et al. (2019), and GPT-2 Radford et al. (2019). These features and label representations are used to compute the transferability metrics required by EMMS, LogME, LEEP, and NLEEP. This comparison setup follows the standard protocol used in EMMS Meng et al. (2023), ensuring consistency and fairness in evaluating model selection quality under limited compute.

### D.1 Evaluation Time Breakdown

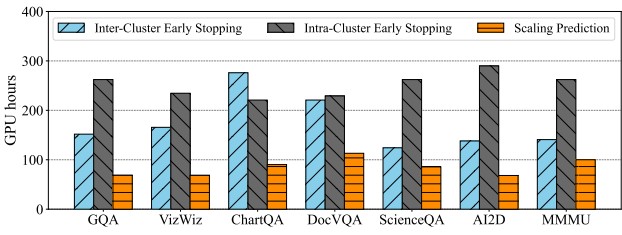

| Dataset | GPU Hours |
|---------|-----------|
| GQA | 6.8 |
| VizWiz | 7.3 |
| ChartQA | 12.3 |
| DocVQA | 9.5 |
| ScienceQA | 5.5 |
| AI2D | 6.0 |
| MMMU | 6.3 |

(a) Total evaluation time breakdown    (b) Cluster computation time

Figure 8: Evaluation time breakdown and cluster computation time.

When evaluating Mordal on different tasks, the total evaluation time required is different. To investigate the differences in time consumption, we analyze the breakdown time for each component of Mordal and present the result in Figure 8. The projector warmup time and CKA computation time is included in the inter-cluster stopping time. For vison encoder, we uses minibatches of size 256 for CKA computation and iterate the target dataset for 5 times. For language model, we use the minibatches of size 32 and iterate the target dataset for 32 times. Sampling is conducted without replacement each time Nguyen et al. (2020b). The cluster computation time (Table 8b) varies across tasks—from 5.5 to 12.3 GPU hours. Generally, tasks with more vision encoder clusters (e.g., ChartQA) require generating more candidate activations, leading to longer cluster computation time. However, this cost is negligible relative to the total evaluation time (typically <3%), due to our two-step clustering strategy and cached activations. The projection warm-up stage is similarly lightweight, requiring only 10 iterations in current setup and completing within one GPU hour for any task.

As shown in Figure 8a, the early stopping stage consists of most of the evaluation time, and the prediction only takes a small part of time. This is because the scaling prediction is only performed for the candidates left after early stopping. The time varies depending on when the model is converged and scaling is observed. The time for inter-cluster and intra-cluster early stopping depends on the number of clusters, controlled by $t_{ve}$ and $t_{llm}$. The cluster is generally less obvious for difficult tasks

(e.g., ChartQA and DocVQA), leading to many clusters with only one candidate. As fewer candidates are eliminated, the total evaluation time increases.

## D.2 Impact of Vision Encoders vs. LLMs

| Model | CLIP | SigLIP | DFN | InternViT |
|---|---|---|---|---|
| Vicuna | 54.8 | 53.0 | 52.7 | 53.4 |
| Mistral | 50.9 | 50.5 | 51.8 | 52.1 |
| Qwen | 61.2 | 65.2 | 64.1 | 61.8 |
| LLaMA3 | 58.2 | 60.1 | 58.8 | 57.0 |

(a) AI2D

| Model | CLIP | SigLIP | DFN | InternViT |
|---|---|---|---|---|
| Vicuna | 41.2 | 44.8 | 34.4 | 30.7 |
| Mistral | 45.1 | 46.9 | 35.6 | 31.3 |
| Qwen | 43.3 | 44.6 | 38.9 | 33.4 |
| LLaMA3 | 37.9 | 38.1 | 35.2 | 32.5 |

(b) VizWiz

Figure 9: Performance across combinations of LLMs and vision encoders on AI2D and VizWiz.

To understand the respective contributions of vision encoders and LLMs, we evaluated $4 \times 4$ combinations of LLMs and vision encoders on two tasks: AI2D and VizWiz. Results are shown in Tables 9a and 9b. On AI2D, the choice of LLM dominates the performance: differences across LLMs (e.g., Qwen vs. Mistral) are significantly larger than differences across vision encoders within each row. On VizWiz, vision encoders dominate—replacing InternViT with SigLIP improves accuracy by 6–15% across all LLMs. These findings are consistent with prior work (Chen et al., 2024a; Wang et al., 2024), which shows that some tasks can be solved without a strong vision encoder.

Mordal accounts for these differences at a coarse granularity. For vision-centric tasks, the clustering stage effectively eliminates underperforming clusters associated with weak vision encoders. Among the remaining candidates—differing mainly in their language backbones—Mordal applies scaling prediction to estimate and rank their performance.

## D.3 Small-scale Models

| Dataset | Top-1 (Grid) | Top-1 (Mordal) | Time (Mordal) | Kendall's $\tau$ |
|---|---|---|---|---|
| GQA | 62.4 (SigLIP-LLaMA) | 62.4 | 189 ($7.7\times$) | 0.707 |
| ChartQA | 12.1 (SigLIP-Gemma) | 12.1 | 217 ($6.7\times$) | 0.621 |
| ScienceQA | 66.8 (SigLIP-LLaMA) | 66.8 | 166 ($8.8\times$) | 0.740 |

Table 7: Small-scale results. Mordal matches grid search in identifying the top-1 model while reducing training costs.

To evaluate scale robustness, we conducted additional experiments using the same seven vision encoders in the paper and smaller LLMs (i.e., Gemma-2-2B, LLaMA-3.2-3B, and Phi-3.5-mini-3B). The results are shown in Table 7. Mordal consistently selects the same top-1 model as full grid search across all tasks, while reducing total training costs by $6.7\times$ to $8.8\times$. However, we observe that Kendall's $\tau$ is lower in the small-model setting. This is likely because smaller models exhibit more inconsistent behavior, making scaling prediction less stable (Zohar et al., 2025). In contrast, larger models tend to demonstrate more predictable scaling behavior, which makes Mordal more effective.

## D.4 Training-free Evaluation

To understand the impact of vision-text alignment on model selection, we consider *training-free* variants of all baselines. In this setting, training-free means that no alignment (i.e., no fine-tuning) is performed before running the model selection baselines. For Mordal, we set the warmup round in candidate clustering to zero so that all methods operate under the same constraint: they only see features from unaligned VLM structures.

Without alignment, the baseline methods (EMMS, LogME, LEEP, and NLEEP) must rely on unaligned image embeddings. As discussed in Section 2, such embeddings are poorly calibrated and LLM cannot reliably interpret them. Although some tasks (e.g., ScienceQA and AI2D) may still perform well by solely relying on the text, the resulting transferability scores are unstable. This mismatch leads to significantly degraded ranking quality and low Kendall $\tau$ as reported in Table 8.

Table 8: Comparison of different model selection methods. Kendall $\tau$ represents the differences of top-performing candidates compared with the groundtruth (i.e., grid search) and *larger $\tau$ is better*.

| Task | Dataset | EMMS | LogME | LEEP | NLEEP | Mordal w/o Warmup |
|------|---------|------|-------|------|-------|-------------------|
| Visual QA | GQA | 0.354 | -0.038 | 0.193 | 0.286 | **0.776** |
| | VizWiz | 0.231 | 0.184 | 0.308 | 0.247 | **0.802** |
| Doc QA | ChartQA | 0.201 | 0.057 | 0.112 | 0.143 | **0.691** |
| | DocVQA | 0.139 | -0.097 | 0.067 | 0.212 | **0.834** |
| Knowledge | ScienceQA | 0.481 | 0.214 | 0.253 | 0.405 | **0.960** |
| | AI2D | 0.323 | 0.137 | 0.169 | 0.359 | **0.855** |
| Multi-discipline | MMMU | 0.242 | 0.148 | 0.299 | 0.163 | **0.737** |

In contrast, Mordal remains relatively robust in the training-free regime. Even without alignment, Mordal relies only on representation *similarity* for candidate clustering: we compute CKA-based similarity between candidates and perform inter-/intra-cluster evaluation on this similarity structure rather than on zero-shot predictions. As a result, the clustering step is much less affected by the lack of alignment, and Mordal still preserves a high weighted Kendall $\tau$ when compared to the fully finetuned grid search ranking. This gap highlights that simply applying training-free transferability metrics to unaligned VLMs is insufficient, whereas Mordal's similarity-driven clustering makes it more robust to the lack of alignment.

### D.5 Q-former Projector

While Mordal's default implementation uses MLP projectors, which are the most prevalent choice in modern VLMs, we investigate the framework's applicability to alternative projector architectures. Specifically, we evaluate Mordal with Q-former projectors Li et al. (2023), which use learnable query tokens and cross-attention mechanisms to bridge vision and language representations. Q-former differs from MLP projectors in several key aspects: (1) it uses a fixed number of learnable query tokens that interact with vision features through cross-attention, (2) it employs self-attention layers to process query tokens, and (3) it outputs a fixed-size representation regardless of input image resolution.

| Language Model | EVA-CLIP-G | CLIP-L |
|----------------|-----------|--------|
| Qwen2-7B | 54.3 | 52.7 |
| Llama-3-8B | 48.8 | 48.7 |
| Mistral-v0.2-7B | 47.8 | 47.2 |
| Vicuna-1.5-7B | 44.9 | 43.4 |

(a) Evaluation Results      (b) EVA-CLIP-G      (c) CLIP-L

Figure 10: Evaluation results of VLMs with four language models and two vision encoders. Tables show the similarity scores between four language models with EVA-CLIP-G and CLIP-L.

We conduct a small-scale experiments on ScienceQA with two vision encoders (EVA-CLIP-G and CLIP-L) and four language models (Qwen2-7B, Llama-3-8B, Mistral-v0.2-7B, and Vicuna-1.5-7B). Note that we use the existing Q-former checkpoints from first stage (i.e., pretraining without involving language model) Dai et al. (2023) and finetune the Q-former projectors with the LLM on a randomly sampled 20% subset of the LLaVA-1.5-Instruction dataset. As presented in Figure 10, CKA remains effective with Q-former projectors. Llama-3-8B achieve similar performance to Mistral-v0.2-7B and have a high CKA score. However, Mordal primarily focuses on MLP projectors and will discuss the choices of projector architectures in Appendix E.

## E Discussion

**Limitations.** Mordal demonstrates promising results in efficiently selecting pretrained models fo VLM with small vision encoders and 7B LLMs under a single-request. However, certain limitations may be addressed to extend its utility and effectiveness. First, while Mordal significantly reduces search time, it does not guarantee perfect top-1 prediction. On challenging tasks, poor clustering or

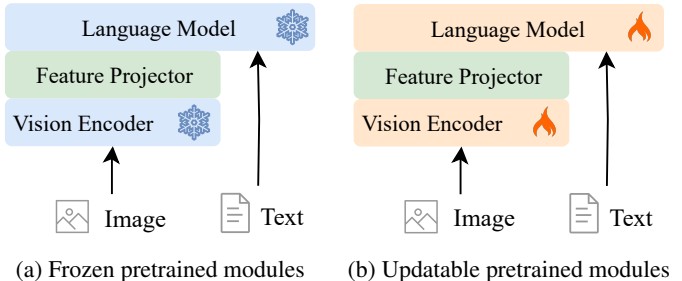

(a) Frozen pretrained modules      (b) Updatable pretrained modules

Figure 11: Alternative alignment approaches for VLM instruction tunning.

early stopping may eliminate strong candidates. Second, Mordal's performance can be sensitive to both dataset and model scale; for instance, scaling prediction becomes less accurate with smaller models, which tend to exhibit higher variance and less predictable behavior. Third, Mordal's performance is sensitive to the a set of hyperparameters. An automated algorithm that balances the cluster number, exploration time and ranking quality would be an important future extension. Despite these limitations, Mordal achieves a balance between efficiency and accuracy. It consistently identifies top-performing models with a speedup of up to $11.6\times$ over grid search, making it a practical solution for real-world VLM selection where exhaustive search is often infeasible.

**VLM alignment.** One must go through an *alignment* process to ensure that the individual components of the VLM are well integrated before using it. Developers integrate pretrained LLMs and visual encoders, training the projector from scratch using visual alignment datasets like VQA Antol et al. (2015) . During the alignment process, pretrained components may remain frozen or be further finetuned during alignment training Liu et al. (2023a). Recently, some proprietary models have employed end-to-end training without using any pretrained models Bai et al. (2023), but it is not common due to the excessive training cost. While Mordal addresses pretrained model selection in a popular VLM setting, it would be interesting to investigate its effectiveness under other VLM structures with and without pretrained components.

**Alternative similarity metrics for candidate clustering.** Mordal uses centered kernel alignment (CKA) as the default similarity metric for candidate clustering. While SVCCA Raghu et al. (2017) and PWCCA Morcos et al. (2018) are popular CCA-based alternatives, replacing CKA with them is practically infeasible for modern VLM embeddings. Standard CCA algorithms scale at least quadratically in the embedding dimension $h$ (e.g., forming or decomposing $h \times h$ covariance matrices), so flattened embeddings with millions of dimensions would require hundreds of gigabytes to terabytes of memory. Moreover, Kornblith et al. (2019) (Theorem 1) show that any similarity index invariant to arbitrary invertible linear transformations (including standard CCA-based indices) becomes degenerate when the representation width exceeds the number of datapoints, making such measures unable to distinguish representations in the over-parameterized regime. By contrast, CKA is invariant only to orthogonal transformations and isotropic scaling, remains well-defined for large $h$, and avoids covariance inversion by operating directly on kernel matrices. Therefore, CKA is the most practical and theoretically sound choice for large-scale candidate clustering in Mordal. An interesting direction for future work is to explore layer-weighted variants of CKA that emphasize layers with higher task relevance or variance, potentially refining candidate space without sacrificing scalability.

**Non-MLP projectors.** We acknowledge the existence of non-MLP projector architectures, such as the Q-former used in BLIP-2 (Li et al., 2023). However, Q-former is being replaced in recent VLMs like BLIP-3 (Xue et al., 2024) due to its limitations. First, Q-former's cross-attention mechanism introduces additional computational overhead. Second, the fixed-size output of Q-former may not capture the full richness of vision representations as effectively as MLP projectors Yao et al. (2024); Huang et al. (2025). Currently, MLP-based projectors are the most prevalent choice in the VLM community due to their simplicity, effectiveness, and compatibility with diverse architectures. They have been shown to generalize well beyond vision-language settings—for instance, in multimodal projection for text, audio, and video in NExT-GPT (Wu et al., 2024). While our current experiments focus on the MLP projector, Mordal is agnostic to the choice of projector architecture, as long as the

projector outputs a fixed embedding. We will expand our discussion in the future version to clarify this point and outline how Mordal could be extended to handle non-MLP and multimodal projectors.

**Extend to smaller and larger models.** Mordal's design and evaluation have focused on small size (i.e., 7B) pretrained models, making it efficient and practical for scenarios with limited computational resources. However, extending Mordal to handle smaller (e.g., 1B) and larger models (e.g., 70B) introduces new challenges. For small models, as shown in Appendix D.3, the similarity measurements and observational scaling law used to speed up evaluation could become less effective with smaller models due to shifts in their feature spaces, potentially reducing the accuracy of candidate selection. For large models, while scaling prediction may becomes more accurate Zohar et al. (2025), the computational overhead associated with larger models increases, requiring more memory and longer processing times for alignment and evaluation. To address these challenges, future iterations of Mordal must incorporate distributed computing frameworks and advanced resource allocation techniques. Adjustments to representation similarity metrics and scaling law formulations will also be essential to maintain the framework's robustness as models vary in size and complexity.

**Similar requests among users.** Mordal's current implementation is designed to optimize pretrained model selection for a single request at a time, which limits its efficiency in handling multiple similar tasks submitted by users. This approach overlooks opportunities to reduce redundant computations when user requests share overlapping requirements, leading to inefficient use of computational resources. By evaluating a shared set of model candidates for grouped tasks, Mordal could eliminate redundant computations and improve throughput. For example, implementing a caching mechanism to store and reuse results for previously evaluated models and tasks could further enhance resource efficiency. Addressing these limitations would enable Mordal to support multi-user environments and dynamic workloads more effectively.

# F Related Work

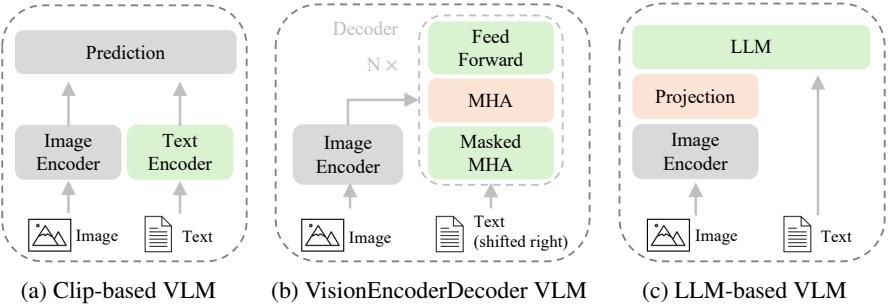

(a) Clip-based VLM     (b) VisionEncoderDecoder VLM     (c) LLM-based VLM

Figure 12: Vision-language models: (a) CLIP-based VLM aligns image and text embeddings via contrastive learning; (b) VisionEncoderDecoder VLM uses an encoder-decoder structure for classification and OCR tasks; (c) LLM-based VLM combines a vision encoder with a language model for multimodal interactions.

**Model selection.** Training-free model selection methods such as EMMS Meng et al. (2023), LogME You et al. (2021), LEEP Nguyen et al. (2020a), and NLEEP Li et al. (2021) assess the transferability of pretrained features without requiring additional finetuning. However, these methods are primarily designed for architectures like CLIP and VisionEncoderDecoder models (as illustrated in Figure 12a and Figure 12b). Other approaches assume that all candidates share the same architecture and differ only in pretraining datasets Tran et al. (2019). LLM-based selection methods Lin et al. (2024) are also not directly applicable to VLMs, as vision and language components are pretrained separately and not jointly aligned. Moreover, most existing techniques are tailored for classification or regression tasks, and struggle with the open-ended nature of multimodal generation. Given the growing number of open-source pretrained models and the unique challenges posed by multimodal alignment, there is a clear need for pretrained model selection approaches specifically designed for LLM-based VLMs (i.e., Figure 12c). Mordal addresses this gap by enabling efficient and alignment-aware selection of pretrained model combinations.

## G    LLM Usage

We used large language models solely to assist in polishing the writing of this paper. No part of the research ideation, experimental design, or analysis relied on LLMs.

