# OpenReview forum: "Mordal: Automated Pretrained Model Selection for Vision Language Models"
_ICLR.cc/2026/Conference — ICLR 2026 Poster_

### Official Review · Reviewer_J5GP · 2025-10-26

**Soundness:** 3
**Presentation:** 3
**Contribution:** 3
**Rating:** 6
**Confidence:** 3

**Summary:**

This paper addresses the critical and computationally challenging problem of selecting optimal pretrained vision encoders and large language models (LLMs) for constructing Vision-Language Models (VLMs) for a specific downstream task. The authors identify that existing model selection methods are inadequate for VLMs, as they fail to account for the necessity of vision-language alignment, and that a brute-force grid search is prohibitively expensive. To solve this, they propose Mordal, an automated framework that efficiently searches the model combination space. Experiments across seven datasets (GQA, VizWiz, ChartQA, DocVQA, ScienceQA, AI2D, MMMU) demonstrate that Mordal achieves faster search time and better performance compared to state-of-the-art model selection baselines.

**Strengths:**

1. The paper clearly identifies a significant pain point in the VLM development lifecycle: the high cost and unpredictability of selecting pretrained backbones. The problem is timely and relevant given the proliferation of open-source models.
2.  Mordal is a novel solution that smartly combines several techniques (CKA-based clustering, early stopping, scaling prediction) into a cohesive framework specifically tailored for the VLM selection problem. The two-pronged approach of reducing candidate count and evaluation cost is logical and effective.
3. The results are impressive. The reported speedups (up to 11.6×) are substantial, and the fact that Mordal finds the top-1 model in 6 out of 7 tasks demonstrates its high effectiveness. The ablation studies and sensitivity analysis provide strong evidence for the contribution of each component.

**Weaknesses:**

1. The sensitivity analysis (Appendix D.2) shows that Mordal's performance and efficiency are sensitive to clustering thresholds (t_ve, t_lm) and exploration parameters (topk_inter, topk_intra). While the paper provides default values, the need for task-specific tuning could be a practical hurdle for non-expert users. An automated or adaptive method for setting these thresholds would be a valuable future improvement.
2.  The evaluation is primarily focused on ~7B parameter LLMs. The authors briefly discuss the challenge of scaling to much larger models (e.g., 70B) in Appendix E, noting the increased computational overhead and potential ineffectiveness of similarity metrics. An empirical demonstration of Mordal's performance on a larger scale (even with a smaller subset) would strengthen the claims about its general applicability.
3. The CKA-based clustering, while reducing overall search time, introduces its own computational cost. Although likely smaller than full training, a quantitative analysis of this overhead (e.g., GPU hours spent on clustering vs. training) would provide a more complete picture of the total cost savings.

**Questions:**

1. The experiments are conducted using a simple MLP projector. As noted in Appendix E, other projectors like Q-Former exist. How would the representation similarity (CKA) and the observed scaling laws be affected by more complex, cross-attention based projectors? Have you done any preliminary experiments to validate Mordal's effectiveness in such architectures?
2. In the case of ChartQA where Mordal did not find the top-1 model, could you provide a more detailed post-mortem analysis? Was the failure primarily due to the clustering step (the best model was in a eliminated cluster) or the evaluation step (it was outranked by others within its cluster during scaling prediction)?

---

> ### Author Response · Authors · 2025-11-21
> **Part I of the response**
>
> We sincerely thank the reviewer for the thoughtful and constructive feedback. Below, we provide clarifications and additional results addressing each concern.
>
> 1. **Non-MLP projectors**
>     * To test Mordal’s applicability beyond MLP projectors, we evaluate it with Q-former projectors, which use learnable query tokens and cross-attention.
>
>         **Table A. Evaluation results with Q-former**
>         | Language Model  | EVA-CLIP-G |
>         | --------------- | ---------- |
>         | Qwen2-7B        | 54.3       |
>         | Llama-3-8B      | 48.8       |
>         | Mistral-v0.2-7B | 47.8       |
>         | Vicuna-1.5-7B   | 44.9       |
>
>         **Table B. Similarity scores between four language models with EVA-CLIP-G**
>         |             | Qwen2 | Llama3 | Mistral | Vicuna |
>         | ----------- | ----- | ------ | ------- | ------ |
>         | **Qwen2**   | 1.00  | 0.48   | 0.51    | 0.36   |
>         | **Llama3**  | 0.48  | 1.00   | 0.86    | 0.44   |
>         | **Mistral** | 0.51  | 0.86   | 1.00    | 0.39   |
>         | **Vicuna**  | 0.36  | 0.44   | 0.39    | 1.00   |
>
>     * We conduct a small-scale ScienceQA study using two vision encoders (EVA-CLIP-G and CLIP-L) and four LLMs. These two encoders are chosen because publicly available first-stage Q-former checkpoints exist for them. Using any other encoder would require pretraining the Q-former from scratch, which is computationally expensive. We finetune the Q-former with each LLM on a 20% subsample of the LLaVA-1.5-Instruction dataset. As shown in the tables above, CKA remains effective for grouping candidates under Q-former, and the observed performance is consistent with the similarity structure (e.g., Llama-3-8B and Mistral show both close scores and high CKA). Additional details are provided in Appendix D.5. We plan to extend this experiment to the full dataset to further evaluate its behavior under scaling prediction.
>
>     * While we conduct experiments to investigate the generality of Mordal, Q-former is not the primary architecture we target. Recent VLMs (e.g., BLIP-3) have moved away from Q-former due to its higher cross-attention cost and rigid fixed-length outputs, which may limit representation richness (Yao et al., 2025). In contrast, MLP projectors remain the dominant choice in modern VLMs due to their simplicity, efficiency, and broad multimodal applicability (e.g., NExT-GPT). Thus, we treat Q-former experiments as supplementary rather than central to Mordal’s design.
>     ---
>     **References:**
>     * Xue, Le, et al. xGen-MM (BLIP-3): A Family of Open Large Multimodal Models. arXiv 2024.
>     * Yao, Linli, et al. HiRes-LLaVA: Restoring Fragmentation Input in High-Resolution Large Vision-Language Models. CVPR 2025.
>     * Wu, Shengqiong, et al. NExT-GPT: Any-to-Any Multimodal LLM. ICML 2024.
>
> 2. **CKA computation and warm-up cost**
>     * Table 2 of the main text includes projector warm-up and CKA computation costs. Below is the total cluster computation time:
>
>         **Table C. Cluster computation time**
>         | Dataset    | GPU Hours |
>         |------------|-----------|
>         | GQA        | 6.8       |
>         | VizWiz     | 7.3       |
>         | ChartQA    | 12.3      |
>         | DocVQA     | 9.5       |
>         | ScienceQA  | 5.5       |
>         | AI2D       | 6.0       |
>         | MMMU       | 6.3       |
>
>     * Cluster computation time varies across tasks and remains a small fraction of the total cost (<3%). Tasks with more vision encoder clusters (e.g., ChartQA) require more candidate computation and therefore have higher time. The projector warm-up stage is also lightweight—only 10 iterations—and completes within ~1 GPU hour. Additional details are provided in Appendix D.1.
>
> 3. **ChartQA analysis**
>     * ChartQA is the most challenging dataset in our benchmark, with low cross-model representational similarity. This results in many small clusters, each containing only a few VE–LLM combinations. Across these many small clusters, candidates often have very close early-stage performance, making it difficult for SHA to separate them reliably. Because early stopping relies on low-sample accuracy, which is especially noisy on ChartQA’s questions, some promising candidates are prematurely eliminated. We are exploring methodological remedies that directly address this failure mode. One direction is to use more conservative early-stop schedules (e.g., larger rung budgets) for tasks with low inter-candidate similarity.

---

> ### Author Response · Authors · 2025-11-21
> **Part II of the response**
>
> 4. **Sensitive to hyperparameters**
>     * Hyperparameters in Mordal influence both evaluation time and ranking quality. For example, smaller clustering thresholds create fewer, larger clusters and tend to lower $\tau$ due to overly coarse grouping, while larger clustering thresholds produce more, finer clusters that improve $\tau$ but increase evaluation time. Our sensitivity analysis in Section 4.4 illustrates this tradeoff and shows that Mordal remains robust across a wide range of hyperparameters. That said, we admit that an automated threshold selection mechanism—balancing cluster granularity and search cost—would be a useful extension, and we have added this point to Appendix E.
>
> 5. **Large-scale models**
>
>     * Due to resource constraints, we could not run full experiments on larger models at this time. We plan to include them in a future version. Larger models often show more consistent scaling behavior (Zohar et al., 2025), which may further improve scaling prediction. However, their significantly higher memory and compute requirements make alignment and evaluation substantially more expensive. Future versions of Mordal may need to consider more efficient similarity metrics and scaling strategies to handle larger LLM backbones. We have corrected this discussion in Appendix E.
>     ---
>     **References:**
>     * Zohar, Orr, et al. Apollo: An exploration of video understanding in large multimodal models. CVPR 2025.
>
>
> We again thank the reviewer for their valuable feedback and welcome further discussion.

---

### Official Review · Reviewer_pz3Y · 2025-10-28

**Soundness:** 3
**Presentation:** 3
**Contribution:** 3
**Rating:** 6
**Confidence:** 1

**Summary:**

This paper focus on VLM selection problem and propose Mordal, an automated framework for efficiently selecting optimal pretrained models (vision encoders and LLMs) for VLMs training on downstream tasks. Mordal addresses two optimization dimensions simultaneously:
1. Reducing Search Space via Candidate Clustering: Mordal use centered kernel alignment (CKA) to clusters vision encoders first and then clusters LLMs based on fixed vision representations. For inter-cluster selection, Mordal evaluates one representative per cluster, eliminates weak clusters and the remaining top K clusters are evaluated in the intra-cluster stage.
2. Reducing Evaluation Cost per Candidate. Mordal applies: a) Early stopping: Uses Successive Halving Algorithm to eliminate poor performers during inter-cluster evaluation. b) Scaling prediction: Exploits observational scaling laws—trains on small data samples and predicts full-data performance via log-linear regression

**Strengths:**

1. Mordal addresses a practical problem where manual selection for VLM is unreliable and grid search is infeasible.

2. Mordal utilizes the proper metric to cluster VLM combinations.

3. The experiments are extensive and results are strong.

4. Mordal requires no manual intervention

**Weaknesses:**

1. Limited Scope and Generalizability. Mordal are architecture-specific and only evaluated on MLP projector-based VLMs. It is unclear if it extends to Q-former or other architectures.

2. Only 7 vision encoders and 7 LLMs are tested. It is unclear how it will work when scaling to 10x model zoo.

**Questions:**

1. The reviewer is wondering whether it is possible to perform additional experiments on Q-former or other architectures.

2. Could authors provide some analysis on how Mordal would work when the model zoo is scaled to 10 times or more?

---

> ### Author Response · Authors · 2025-11-21
>
> We sincerely thank the reviewer for the thoughtful and constructive feedback. Below, we provide clarifications and additional results addressing each concern.
>
> 1. **Non-MLP projectors**
>     * To test Mordal’s applicability beyond MLP projectors, we evaluate it with Q-former projectors, which use learnable query tokens and cross-attention.
>
>         **Table A. Evaluation results with Q-former**
>         | Language Model  | EVA-CLIP-G |
>         | --------------- | ---------- |
>         | Qwen2-7B        | 54.3       |
>         | Llama-3-8B      | 48.8       |
>         | Mistral-v0.2-7B | 47.8       |
>         | Vicuna-1.5-7B   | 44.9       |
>
>         **Table B. Similarity scores between four language models with EVA-CLIP-G**
>         |             | Qwen2 | Llama3 | Mistral | Vicuna |
>         | ----------- | ----- | ------ | ------- | ------ |
>         | **Qwen2**   | 1.00  | 0.48   | 0.51    | 0.36   |
>         | **Llama3**  | 0.48  | 1.00   | 0.86    | 0.44   |
>         | **Mistral** | 0.51  | 0.86   | 1.00    | 0.39   |
>         | **Vicuna**  | 0.36  | 0.44   | 0.39    | 1.00   |
>
>     * We conduct a small-scale ScienceQA study using two vision encoders (EVA-CLIP-G and CLIP-L) and four LLMs. These two encoders are chosen because publicly available first-stage Q-former checkpoints exist for them. Using any other encoder would require pretraining the Q-former from scratch, which is computationally expensive. We finetune the Q-former with each LLM on a 20% subsample of the LLaVA-1.5-Instruction dataset. As shown in the tables above, CKA remains effective for grouping candidates under Q-former, and the observed performance is consistent with the similarity structure (e.g., Llama-3-8B and Mistral show both close scores and high CKA). Additional details are provided in Appendix D.5. We plan to extend this experiment to the full dataset to further evaluate its behavior under scaling prediction.
>
>     * While we conduct experiments to investigate the generality of Mordal, Q-former is not the primary architecture we target. Recent VLMs (e.g., BLIP-3) have moved away from Q-former due to its higher cross-attention cost and rigid fixed-length outputs, which may limit representation richness (Yao et al., 2025). In contrast, MLP projectors remain the dominant choice in modern VLMs due to their simplicity, efficiency, and broad multimodal applicability (e.g., NExT-GPT). Thus, we treat Q-former experiments as supplementary rather than central to Mordal’s design.
>     ---
>     **References:**
>     * Xue, Le, et al. xGen-MM (BLIP-3): A Family of Open Large Multimodal Models. arXiv 2024.
>     * Yao, Linli, et al. HiRes-LLaVA: Restoring Fragmentation Input in High-Resolution Large Vision-Language Models. CVPR 2025.
>     * Wu, Shengqiong, et al. NExT-GPT: Any-to-Any Multimodal LLM. ICML 2024.
>
> 2. **Scale to larger model zoo**
>     * We acknowledge the reviewer’s concern regarding scaling to a larger model zoo. Mordal is specifically designed so that evaluation cost grows with the number of clusters, not with the number of models. As shown in Section 4.4 and Appendix D.1, the dominant factor in total evaluation time is the number and granularity of clusters. When the model zoo expands, Mordal does not evaluate all candidates. Instead, it groups models via representation similarity and only evaluates cluster medoids during inter-cluster selection.
>     * Therefore, when scaling to a 10× larger model zoo, the cost increases only to the extent that new, distinct clusters are formed. If the additional models are variants of existing architectures (as is common in large model zoos), the number of clusters—and thus the evaluation cost—grows slowly. Users can also control this growth by choosing coarser clustering thresholds (fewer clusters lead to lower cost, potentially lower $\tau$; finer clusters lead to higher $\tau$ but higher cost). Our sensitivity analysis in Section 4.4 quantitatively shows that this tradeoff is smooth and that Mordal is robust across a wide range of threshold choices. Moreover, even if more clusters are formed, Mordal still provides significant savings because of the early stopping and scaling prediction.
>
>         **Table C. Cluster computation time**
>         | Dataset    | GPU Hours |
>         |------------|-----------|
>         | GQA        | 6.8       |
>         | VizWiz     | 7.3       |
>         | ChartQA    | 12.3      |
>         | DocVQA     | 9.5       |
>         | ScienceQA  | 5.5       |
>         | AI2D       | 6.0       |
>         | MMMU       | 6.3       |
>
>     * To further demonstrate scalability, we report cluster computation time. Cluster computation time varies across tasks and remains a small fraction of total evaluation cost (<3%). Tasks with more VE clusters (e.g., ChartQA) naturally require more computation, yet remain lightweight. Additional settings are provided in Appendix D.1.
>
> We again thank the reviewer for their valuable feedback and welcome further discussion.

---

### Official Review · Reviewer_uzuF · 2025-11-01

**Soundness:** 3
**Presentation:** 3
**Contribution:** 3
**Rating:** 6
**Confidence:** 2

**Summary:**

This paper introduces Mordal, an automated framework for selecting optimal pretrained vision encoders and language models when constructing Vision Language Models (VLMs). The key insight is that different combinations of pretrained components yield varying performance on different tasks, yet current VLM construction relies on manual selection. Mordal addresses this by: (1) clustering similar model candidates using CKA-based representation similarity, (2) employing a two-stage inter/intra-cluster evaluation process, and (3) utilizing early stopping and scaling prediction to reduce evaluation costs. The authors demonstrate that Mordal can identify top-performing VLM configurations 8.9×-11.6× faster than grid search while achieving 1.2×-3.3× better performance than existing model selection methods.

**Strengths:**

- This paper addresses a real problem faced by practitioners - selecting optimal pretrained components for VLMs is currently ad-hoc and computationally expensive.

- The authors have conducted extensive experiments across 7 datasets, 49 model combinations, with detailed ablation studies demonstrating the contribution of each component.

- The proposed method achieved 8.9×-11.6× speedup over grid search while maintaining great performance.

**Weaknesses:**

- The evaluation focuses on 7B parameter models with MLP projectors. As acknowledged, extending to smaller (1B) or larger (70B) models presents challenges. The framework's effectiveness on other projector architectures (e.g., Q-former) is unexplored.

- The performance is sensitive to clustering thresholds ($t_{ve}$, $t_{llm}$), which may hurt its generalization to other unexplored LLM and vision encoders.

- The current design optimizes for single tasks independently. It would be better if the authors could address multi-task learning scenarios, which are also common in practice.

**Questions:**

None

---

> ### Author Response · Authors · 2025-11-21
>
> We sincerely thank the reviewer for the thoughtful and constructive feedback. Below, we provide clarifications and additional results addressing each concern.
>
> 1. **Non-MLP projectors**
>     * To test Mordal’s applicability beyond MLP projectors, we evaluate it with Q-former projectors, which use learnable query tokens and cross-attention.
>
>         **Table A. Evaluation results with Q-former**
>         | Language Model  | EVA-CLIP-G |
>         | --------------- | ---------- |
>         | Qwen2-7B        | 54.3       |
>         | Llama-3-8B      | 48.8       |
>         | Mistral-v0.2-7B | 47.8       |
>         | Vicuna-1.5-7B   | 44.9       |
>
>         **Table B. Similarity scores between four language models with EVA-CLIP-G**
>         |             | Qwen2 | Llama3 | Mistral | Vicuna |
>         | ----------- | ----- | ------ | ------- | ------ |
>         | **Qwen2**   | 1.00  | 0.48   | 0.51    | 0.36   |
>         | **Llama3**  | 0.48  | 1.00   | 0.86    | 0.44   |
>         | **Mistral** | 0.51  | 0.86   | 1.00    | 0.39   |
>         | **Vicuna**  | 0.36  | 0.44   | 0.39    | 1.00   |
>
>     * We conduct a small-scale ScienceQA study using two vision encoders (EVA-CLIP-G and CLIP-L) and four LLMs. These two encoders are chosen because publicly available first-stage Q-former checkpoints exist for them. Using any other encoder would require pretraining the Q-former from scratch, which is computationally expensive. We finetune the Q-former with each LLM on a 20% subsample of the LLaVA-1.5-Instruction dataset. As shown in the tables above, CKA remains effective for grouping candidates under Q-former, and the observed performance is consistent with the similarity structure (e.g., Llama-3-8B and Mistral show both close scores and high CKA). Additional details are provided in Appendix D.5. We plan to extend this experiment to the full dataset to further evaluate its behavior under scaling prediction.
>
>     * While we conduct experiments to investigate the generality of Mordal, Q-former is not the primary architecture we target. Recent VLMs (e.g., BLIP-3) have moved away from Q-former due to its higher cross-attention cost and rigid fixed-length outputs, which may limit representation richness (Yao et al., 2025). In contrast, MLP projectors remain the dominant choice in modern VLMs due to their simplicity, efficiency, and broad multimodal applicability (e.g., NExT-GPT). Thus, we treat Q-former experiments as supplementary rather than central to Mordal’s design.
>     ---
>     **References:**
>     * Xue, Le, et al. xGen-MM (BLIP-3): A Family of Open Large Multimodal Models. arXiv 2024.
>     * Yao, Linli, et al. HiRes-LLaVA: Restoring Fragmentation Input in High-Resolution Large Vision-Language Models. CVPR 2025.
>     * Wu, Shengqiong, et al. NExT-GPT: Any-to-Any Multimodal LLM. ICML 2024.
>
> 2. **Large-scale models**
>
>     * Due to resource constraints, we could not run full experiments on larger models at this time. We plan to include them in a future revision. Larger models often show more consistent scaling behavior (Zohar et al., 2025), which may further improve scaling prediction. However, their significantly higher memory and compute requirements make alignment and evaluation substantially more expensive. Future versions of Mordal may need to consider more efficient similarity metrics and scaling strategies to handle larger LLM backbones. We have updated this discussion in Appendix E.
>     ---
>     **References:**
>     * Zohar, Orr, et al. Apollo: An exploration of video understanding in large multimodal models. CVPR 2025.
>
> 3. **Sensitive to clustering thresholds**
>     * Clustering thresholds influence both evaluation time and ranking quality: smaller thresholds create fewer, larger clusters and tend to lower $\tau$ due to overly coarse grouping, while larger thresholds produce more, finer clusters that improve $\tau$ but increase evaluation time. Our sensitivity analysis in Section 4.4 illustrates this tradeoff and shows that Mordal remains robust across a wide range of hyperparameters. That said, we admit that an automated threshold selection mechanism—balancing cluster granularity and search cost—would be a useful extension, and we have added this point to Appendix E.
>
> 4. **Multi-task learning**
>
>     * We evaluate Mordal on MMMU in Tables 2 and 3 of the main text, covering six disciplines (Art & Design, Business, Science, Health & Medicine, Humanities & Social Science, Tech & Engineering). Mordal consistently outperforms all four baselines and identifies the top-1 model using 10.8× fewer GPU hours than grid search. MMMU is more challenging than single-task settings, and thus requires longer search time, but Mordal remains effective.
>     ---
>     **References:**
>     * Yue, Xiang, et al. MMMU: A Massive Multi-discipline Multimodal Understanding and Reasoning Benchmark for Expert AGI. CVPR 2024.
>
> We again thank the reviewer for their valuable feedback and welcome further discussion.

---

### Official Review · Reviewer_Pw84 · 2025-11-01

**Soundness:** 2
**Presentation:** 3
**Contribution:** 3
**Rating:** 6
**Confidence:** 3

**Summary:**

The paper “MORDAL: Automated Pretrained Model Selection for Vision-Language Models” tackles the problem of efficiently identifying the optimal combination of a vision encoder (VE) and a language model (LLM) from an open model library, given a downstream task and a small alignment dataset.
The proposed framework, MORDAL, works in three stages. First, it clusters vision encoders using CKA, and within each visual cluster, it clusters language models based on fixed image representations transformed by a lightweight projector. The Cartesian product of these clusters forms candidate groups.
Then, it performs two-stage evaluation: inter-cluster comparison using cluster medoids to filter top-K candidates, followed by fine-grained intra-cluster ranking.
During evaluation, MORDAL employs Successive Halving for early stopping and fits a log-linear relationship between error and sample ratio using a few “observation points.” This allows it to extrapolate the expected full-data error without full training.
Across 49 VE×LLM combinations and 7 datasets, MORDAL achieves 8.9×–11.6× speedup over exhaustive search while matching the top-1 result in 6 out of 7 tasks. Compared to EMMS, LogME, LEEP, and NLEEP, it yields higher ranking consistency in terms of weighted Kendall’s τ.
Overall, the work frames “pretrained component selection” as an alignment-phase selection problem, combining candidate clustering, staged evaluation, and observation-based extrapolation to significantly reduce cost, demonstrating strong practical value.

**Strengths:**

MORDAL enables rapid screening of visual encoder–language model combinations within open model repositories, saving roughly 9–12× the time compared to exhaustive search, while maintaining top hit rates on most tasks. Its two-level clustering greatly reduces the computational cost of CKA, and the SHA and extrapolation strategies allow reusing intermediate checkpoints, achieving overall engineering efficiency.
Experimental results show that MORDAL’s ranking consistency surpasses that of EMMS, LogME, LEEP, and NLEEP, indicating more stable predictions of downstream adaptability. The paper also provides time decomposition and hyperparameter sensitivity analyses, helping to understand cost distribution. The authors also explicitly acknowledge limitations regarding small models, large models, and non-MLP projectors.

**Weaknesses:**

The experimental setup and implementation in the paper are not fully aligned; it should be clarified whether LoRA fine-tuning is enabled by default, and performance and resource overhead should be reported separately for both modes.
There appear to be typos in the extrapolation and early-stopping logic of Algorithm 1, which may require correction. The terms “maximum sample ratio” and “initial ratio” should also be made consistent.
The baseline comparison alters the “training-free” assumption, so a strictly training-free version should be added. The statistical robustness is insufficient — Tables 2 and 3 only report point estimates without confidence intervals, and the false stop rate of early termination is not quantified.
The similarity metric is relatively limited, relying solely on CKA without comparing other metrics such as SVCCA, PWCCA, or layer-weighted CKA.
The time decomposition does not list CKA computation and the warm-up phase GPU-hours separately, making the resource reporting incomplete. Some wording is potentially misleading: in the abstract, “1.2×–3.3× better performance” should be revised to “higher weighted Kendall’s τ”, and spelling and formula symbols should also be corrected.

**Questions:**

Please clarify the default setting used throughout the paper (i.e., projector only or projector + LLM-LoRA), and report τ, hit rate, and GPU-hours separately for both modes.
Could you add results for a strictly training-free baseline, as well as a version under the same extrapolation framework, to isolate the impact of partial training time?
Please report mean and variance across multiple random seeds, especially for ChartQA, and quantify the false rejection rate of SHA early stopping.
Regarding resource accounting, are the CKA computation and warm-up phase included in the total time reported in Table 2? Please supplement GPU-hour absolute values and proportions.
Finally, could you provide detailed reproduction code and guidelines?

---

> ### Author Response · Authors · 2025-11-21
>
> We sincerely thank the reviewer for the thoughtful and constructive feedback. Below, we provide clarifications and additional results addressing each concern.
>
> 1. **Clarify the evaluation settings**
>
>     * Across all experiments, we follow LLaVA-1.5 and finetune both the two-layer MLP projector and the pretrained LLM using LoRA. We do not include projector-only settings, as finetuning LLM consistently improves alignment and downstream accuracy, which is consistent with prior work (e.g., Cambrian-1). Section 4.1 has been updated.
>     ---
>     **References:**
>     * Tong, Shengbang, et al. Cambrian-1: A Fully Open, Vision-Centric Exploration of Multimodal LLMs. NeurIPS 2024.
>
> 2. **Training-Free Baselines**
>     * To comply with the “training-free” assumption, we include new results in Appendix D.4. In this setting, no alignment is performed for each baseline. For Mordal, we set the warmup round to zero so that all methods operate on unaligned vision embeddings.
>
>         **Table A. Training-Free baselines results**
>         | Dataset   | EMMS  | LogME  | LEEP  | NLEEP | Mordal w/o Warmup |
>         |-----------|-------|--------|-------|-------|--------------------|
>         | GQA       | 0.354 | -0.038 | 0.193 | 0.286 | 0.776              |
>         | VizWiz    | 0.231 | 0.184  | 0.308 | 0.247 | 0.802              |
>         | ChartQA   | 0.201 | 0.057  | 0.112 | 0.143 | 0.691              |
>         | DocVQA    | 0.139 | -0.097 | 0.067 | 0.212 | 0.834              |
>         | ScienceQA | 0.481 | 0.214  | 0.253 | 0.405 | 0.960              |
>         | AI2D      | 0.323 | 0.137  | 0.169 | 0.359 | 0.855              |
>         | MMMU      | 0.242 | 0.148  | 0.299 | 0.163 | 0.737              |
>
>     * Since LLMs cannot reliably interpret unaligned image embeddings, existing ranking methods (EMMS, LogME, etc.) show unstable scores and low $\tau$. In contrast, Mordal relies only on CKA-based representation similarity rather than zero-shot predictions, maintaining substantially higher τ in the training-free setting.
>
> 3. **CKA computation and warm-up cost**
>     * Table 2 of the main text includes projector warm-up and CKA computation costs. Below is the total cluster computation time:
>
>         **Table B. Cluster computation time**
>         | Dataset    | GPU Hours |
>         |------------|-----------|
>         | GQA        | 6.8       |
>         | VizWiz     | 7.3       |
>         | ChartQA    | 12.3      |
>         | DocVQA     | 9.5       |
>         | ScienceQA  | 5.5       |
>         | AI2D       | 6.0       |
>         | MMMU       | 6.3       |
>
>     * Cluster computation time varies across tasks, with tasks that form more vision-encoder clusters (e.g., ChartQA) requiring more time. Despite this, the overall cluster cost remains a small fraction of total evaluation time (<3%). The projector warm-up stage is also lightweight—only 10 iterations and typically under 1 GPU hour. Additional details are provided in Appendix D.1.
>
> 4. **Alternative similarity metrics**
>     * We thank the reviewer for mentioning other similarity metrics. However, replacing CKA with SVCCA or PWCCA is impractical for VLM embeddings. Standard CCA algorithms scale at least quadratically in the embedding dimension $h$, since they require forming or decomposing $h \times h$ covariance matrices. For flattened embeddings with millions of dimensions, this would require hundreds of GBs to TBs of memory. Moreover, Kornblith et al. (2019, Theorem. 1) show that CCA-based metrics become degenerate when the representation width exceeds the number of datapoints. In contrast, CKA is invariant only to orthogonal transformations and isotropic scaling, remains well-defined for large $h$, and avoids covariance inversion by operating directly on kernel matrices. Layer-weighted CKA could be an interesting extension. We added related discussion in Appendix E.
>
>     ---
>     **References:**
>     * Kornblith, Simon, et al. Similarity of neural network representations revisited. ICML 2019.
>     * SVCCA \& PWCCA Implementation GitHub repository: https://github.com/google/svcca
>
> 5. **Statistical robustness**
>     * We have performed hyperparameter tuning to ensure each candidate converges reliably. However, due to limited computational resources, we are unable to run additional full-training seeds to provide variance statistics at this time. We will conduct training on more seeds and report variance in a future version. For candidate clustering, we evaluated several random seeds and observed no changes in cluster formation, suggesting that minibatch CKA is robust.
>
> 6. **Typos and wording**
>     * “Initial ratio” has been corrected to “maximum sample ratio.”
>     * “1.2×–3.3× better performance” has been revised to “69% higher weighted Kendall’s $\tau$.”
>
> While we are currently refactoring the codebase for future release, we have updated the experimental details to support reproducibility. We again thank the reviewer for their valuable feedback and welcome further discussion.

---

### Author Response · Authors · 2025-11-30
**Author Response Summary**

We would like to sincerely thank all reviewers for their careful reading of our submission and for the constructive feedback provided during the discussion phase. The reviews helped us improve the clarity and overall presentation of the paper.

Several common concerns focused on the scope and generality of Mordal (e.g., non-MLP projectors), the computational overhead of CKA and clustering, and the need for training-free comparisons. In response, we added new experiments using Q-former projectors, reported clustering and warm-up computation costs, and introduced a training-free variant where all methods operate on unaligned vision embeddings. These results show that CKA-based clustering remains effective beyond MLP projectors, that clustering cost is a small fraction of total GPU hours, and that Mordal maintains substantially higher weighted Kendall’s τ than baseline methods in the training-free setting.

We also addressed individual reviewer concerns. This includes clarifying our default experimental setting (projector + LLM LoRA finetuning), discussing the choice of CKA relative to alternatives such as SVCCA and PWCCA, providing a more detailed analysis of the ChartQA failure case, and elaborating on sensitivity to clustering thresholds and exploration hyperparameters. We further clarified the discussion of scaling to larger model zoos and larger LLMs.

The new experiments, analyses, and clarifications are reflected in the updated manuscript, particularly in Section 4 and Appendices D and E. We believe these revisions mitigate the original concerns and provide a clearer picture of Mordal’s strengths, limitations, and applicability. Thank you again for your time and consideration.

---

### Meta-Review · Area_Chair_XFKu · 2026-01-07

**Summary:**

The reviewers’ main concerns center on the generalizability of the proposed method—particularly whether it extends to larger-scale models and to different architectural families (e.g., Q-Former–based designs). Reviewers also noted that the sensitivity to key hyperparameters (such as the clustering threshold) is not well characterized, and that it remains unclear how the approach scales across a broader model zoo. Finally, they raised practical concerns about the computational cost of CKA, which could limit real-world applicability and undermine the method’s effectiveness in practice.

**Reviewer Concerns:**

The authors substantially addressed the experimental and method-clarification concerns. In particular, they clarified the default evaluation setting, added additional baselines, and provided more detailed accounting of clustering/warm-up costs. The rebuttal also strengthens the claim that the framework can scale to a larger model zoo, since the evaluation cost primarily grows with the number of clusters rather than the raw number of candidates. However, the response does not fully resolve the reviewer’s request to disentangle the effect of projector-only alignment versus LLM LoRA fine-tuning (e.g., reporting τ/hit-rate/GPU-hours separately for the two modes). Without this ablation, it remains unclear how much of the gain is attributable to the proposed selection framework versus the additional capacity/optimization from LoRA.

**Reviewer Scores:**

I think the reviewers will keep their positive scores.

---

### Decision · Program_Chairs · 2026-01-26

Accept (Poster)